# Impact of point-of-care ultrasound and routine third trimester ultrasound on undiagnosed breech presentation and perinatal outcomes: An observational multicentre cohort study

**Samantha Knights**[1], **Smriti Prasad**[2], **Erkan Kalafat**[3,4], **Anahita Dadali**[2], **Pam Sizer**[1], **Francoise Harlow**[1], **Asma Khalil**[2,5,6] *

1 Department of Obstetrics and Gynaecology, Norfolk and Norwich University Hospitals NHS Foundation Trust, Norwich, United Kingdom, 2 Fetal Medicine Unit, St George's University Hospitals NHS Foundation Trust, London, United Kingdom, 3 Department of Statistics, Middle East Technical University, Faculty of Arts and Sciences, Ankara, Turkey, 4 Department of Obstetrics and Gynaecology, Koc University, School of Medicine, Istanbul, Turkey, 5 Vascular Biology Research Centre, Molecular and Clinical Sciences Research Institute, St George's University of London, London, United Kingdom, 6 Fetal Medicine Unit, Liverpool Women's Hospital, Liverpool, United Kingdom

☯ These authors contributed equally to this work.

* akhalil@sgul.ac.uk

**Data Availability Statement:** Data cannot be shared publicly because consent was not obtained from women; permission for sharing data was not sought as part of ethical approval. Data is only available following approval from Research Ethics

## Abstract

### Background

Accurate knowledge of fetal presentation at term is vital for optimal antenatal and intrapartum care. The primary objective was to compare the impact of routine third trimester ultrasound or point-of-care ultrasound (POCUS) with standard antenatal care, on the incidence of overall and proportion of all term breech presentations that were undiagnosed at term, and on the related adverse perinatal outcomes.

### Methods and findings

This was a retrospective multicentre cohort study where we included data from St. George's (SGH) and Norfolk and Norwich University Hospitals (NNUH). Pregnancies were grouped according to whether they received routine third trimester scan (SGH) or POCUS (NNUH). Women with multiple pregnancy, preterm birth prior to 37 weeks, congenital abnormality, and those undergoing planned cesarean section for breech presentation were excluded. Undiagnosed breech presentation was defined as follows: (a) women presenting in labour or with ruptured membranes at term subsequently discovered to have a breech presentation; and (b) women attending for induction of labour at term found to have a breech presentation before induction. The primary outcome was the proportion of all term breech presentations that were undiagnosed. The secondary outcomes included mode of birth, gestational age at birth, birth weight, incidence of emergency cesarean section, and the following neonatal adverse outcomes: Apgar score <7 at 5 minutes, unexpected neonatal unit

Committee and Confidentiality Advisory Group. Enquiries and requests should be made to the the Research Governance and Delivery team at St George's University of London (sgulREC@sgul.ac. uk).

**Funding:** The author(s) received no specific funding for this work.

**Abbreviations:** BAME, black, Asian, and minority ethnic; BMI, body mass index; CrI, credible intervals; ECV, external cephalic version; HIE, hypoxic ischemic encephalopathy; HRA, Health Research Authority; HTA, Health Technology Assessment; IMD, index of multiple deprivation; NICE, National Institute for Health and Care Excellence; NIHR, National Institute for Health Research; NNU, neonatal unit; NNUH, Norfolk and Norwich University Hospital; NSC, National Screening Committee; POCUS, point-of-care ultrasound; RR, risk ratio; SGH, St. George's Hospital.

(NNU) admission, hypoxic ischemic encephalopathy (HIE), and perinatal mortality (including stillbirths and early neonatal deaths). We employed a Bayesian approach using informative priors from a previous similar study; updating their estimates (prior) with our own data (likelihood). The association of undiagnosed breech presentation at birth with adverse perinatal outcomes was analyzed with Bayesian log-binomial regression models. All analyses were conducted using R for Statistical Software (v.4.2.0).

Before and after the implementation of routine third trimester scan or POCUS, there were 16,777 and 7,351 births in SGH and 5,119 and 4,575 in NNUH, respectively. The rate of breech presentation in labour was consistent across all groups (3% to 4%). In the SGH cohort, the percentage of all term breech presentations that were undiagnosed was 14.2% (82/578) before (years 2016 to 2020) and 2.8% (7/251) after (year 2020 to 2021) the implementation of universal screening ($p < 0.001$). Similarly, in the NNUH cohort, the percentage of all term breech presentations that were undiagnosed was 16.2% (27/167) before (year 2015) and 3.5% (5/142) after (year 2020 to 2021) the implementation of universal POCUS screening ($p < 0.001$). Bayesian regression analysis with informative priors showed that the rate of undiagnosed breech was 71% lower after the implementation of universal ultrasound (RR, 0.29; 95% CrI 0.20, 0.38) with a posterior probability greater than 99.9%. Among the pregnancies with breech presentation, there was also a very high probability (>99.9%) of reduced rate of low Apgar score (<7) at 5 minutes by 77% (RR, 0.23; 95% CrI 0.14, 0.38). There was moderate to high probability (posterior probability: 89.5% and 85.1%, respectively) of a reduction of HIE (RR, 0.32; 95% CrI 0.0.05, 1.77) and extended perinatal mortality rates (RR, 0.21; 95% CrI 0.01, 3.00). Using informative priors, the proportion of all term breech presentations that were undiagnosed was 69% lower after the initiation of universal POCUS (RR, 0.31; 95% CrI 0.21, 0.45) with a posterior probability greater of 99.9%. There was also a very high probability (99.5%) of a reduced rate of low Apgar score (<7) at 5 minutes by 40% (RR, 0.60; 95% CrI 0.39, 0.88). We do not have reliable data on number of facility-based ultrasound scans via the standard antenatal referral pathway or external cephalic versions (ECVs) performed during the study period.

## Conclusions

In our study, we observed that both a policy of routine facility-based third trimester ultrasound or POCUS are associated with a reduction in the proportion of term breech presentations that were undiagnosed, with an improvement in neonatal outcomes. The findings from our study support the policy of third trimester ultrasound scan for fetal presentation. Future studies should focus on exploring the cost-effectiveness of POCUS for fetal presentation.

### Author summary

#### Why was this study done?

- Accurate knowledge of fetal presentation is essential for optimal care during pregnancy and birth. Vaginal breech delivery is associated with adverse maternal and perinatal outcomes.

- Abdominal palpation has poor sensitivity (50% to 70%) for determination of fetal presentation.

- The role of a routine third ultrasound assessment of fetal presentation has been reported but the impact on neonatal outcomes is yet to be determined.

- There are limited reports on antenatal use of handheld point-of-care ultrasound (POCUS) for the determination of fetal presentation, but the impact of their systematic use for this purpose is largely unknown.

## What did the researchers do and find?

- We analysed 2 cohorts of pregnant women from 2 large teaching hospitals in the United Kingdom where a policy of routine third trimester ultrasound or POCUS has been implemented.

- We studied the impact of routine third trimester ultrasound or POCUS on the percentage of all term breech presentations that were undiagnosed and adverse neonatal outcomes, in pre- and post-screening epochs.

- Due to the rarity of adverse outcomes, we employed Bayesian regression analysis with informative priors. This statistical tool permits updating previous findings with new data to generate new evidence.

- We found that the incidence of all term breech presentations that were undiagnosed reduced drastically in the post-screening epoch following the implementation of either a third trimester ultrasound (decreased from 14.2% to 2.8%) or POCUS (decreased from 16.2% to 3.5%). There was an associated improvement in neonatal outcomes.

## What do these findings mean?

- Our findings imply that a policy of either a third trimester ultrasound by sonographers or POCUS by trained midwives was effective in reducing the proportion of all term breech presentations at the time of birth that were undiagnosed and associated neonatal complications.

- Cost-effectiveness of POCUS needs to be explored further for feasibility of implementation on a wider scale for assessment of fetal presentation at term.

## Introduction

The incidence of breech presentation at term is 3% to 4% [1]. Breech vaginal birth is associated with an increase in both perinatal mortality and morbidity as well as maternal morbidity [2–7]. Correct knowledge of fetal presentation at term is essential for providing optimum antepartum and intrapartum care. Women with breech presentation at term can be effectively counselled about their options—external cephalic version (ECV), planned vaginal birth, or elective cesarean birth—with their inherent risks and perceived benefits [1]. There is substantial evidence that clinical examination is not accurate enough for determination of fetal presentation, with unacceptably high rates of missed breech/noncephalic presentations at term [8,9].

There are 2 modalities to screen for fetal presentation at term, each with its advantages and disadvantages: routine third trimester ultrasound or point-of-care/portable ultrasound (POCUS). Currently, routine third trimester ultrasound is not recommended by the United Kingdom National Institute for Health and Care Excellence (NICE) in low-risk pregnancies due to insufficient clinical and cost-effectiveness evidence [10,11]. In the UK, the current practice is to perform an early pregnancy risk assessment followed by referral pathways for low-risk and high-risk women. These risks relate to maternal, fetal, and placental pathology but are unrelated to the risk of breech presentation at term. Women deemed to be at high risk are referred for an ultrasound scan at 28 weeks' gestation for fetal biometry with or without additional follow-up ultrasound scans. Low-risk women are followed up with clinical assessment (serial measurement of symphysio-fundal height) and referred for third trimester ultrasound if fetal growth restriction is suspected or if it is difficult to perform clinical examination, as in women with high body mass index (BMI), multiple pregnancy, or multiple uterine fibroids, or there is clinical suspicion of noncephalic fetal presentation at term [12–14]. Emerging data from observational studies and a systematic review indicate that it is feasible to accurately diagnose fetal presentation at term by third trimester ultrasound, thereby reducing the proportion of all term breech presentations that were undiagnosed at the time of labour and birth [15–18]. The clinical end point of any study of the diagnosis of breech presentation at term would be an improvement in neonatal outcomes, associated with reduction in incidence of undiagnosed breech. Hitherto published literature, however, could not demonstrate a translation of increased antenatal diagnosis of breech presentation into a statistically significant improvement in neonatal outcomes, most likely owing to the rarity of adverse outcomes.

Most of the data on the use of POCUS in antenatal settings are from low-resource settings where there is inadequate access to ultrasound owing to both material and physical constraints; hence, the focus is on task-shifting of obstetric ultrasound from sonographers to primary care providers [19,20]. A recently published review reported improved diagnostic accuracy with POCUS compared to clinical examination only, for high-risk obstetric conditions including fetal malpresentation, albeit studies were heterogeneous and referred to varying standards [21]. The Society of Obstetricians and Gynaecologists of Canada identifies POCUS as a useful modality for timely determination of fetal presentation [22]. A retrospective criterion-based audit performed in one of our study hospitals demonstrated that the use of POCUS by midwives in the antenatal ward/labour ward was associated with identification of previously unrecognized breech presentation, thereby preventing inappropriate induction of labour [23]. A recent validation study of POCUS in obstetric care showed near perfect agreement for assessment of fetal presentation [95.6% agreement, Kappa −0.887, 95% CI (0.78 to 0.99)] when compared to routine ultrasound [24]. There is, however, scanty literature on the diagnostic accuracy of POCUS in antenatal care settings for assessment of fetal presentation, compared to standard antenatal care, i.e., routine abdominal palpation, with referral for ultrasound when there is clinical suspicion of breech presentation.

In our study, we aimed to compare the impact of routine third trimester ultrasound or POCUS with standard antenatal care, on the incidence of overall and proportion of all term breech presentations that were undiagnosed at term, and on the related adverse perinatal outcomes.

## Methods

The study included data from St. George's University Hospital NHS Foundation Trust (SGH) and Norfolk and Norwich University Hospital NHS Foundation Trust (NNUH). For both centres, pregnancies were grouped according to whether they received routine third trimester scan (SGH) or POCUS (NNUH).

## Routine third trimester scan cohort

We included a cohort of pregnant women who gave birth between 4 April 2016 and 30 September 2021, at SGH, a large teaching hospital in South West London. The chosen starting point was the date when birth records were first systematically entered into the current electronic database. At SGH, a policy of routine third trimester (at 36 weeks) ultrasound scan by sonographers for all pregnant women has been implemented since January 2020; this includes assessment of fetal biometry, umbilical and middle cerebral artery Doppler, placental localization, amniotic fluid volume, and fetal presentation. Following a diagnosis of breech presentation during the ultrasound scan, women are counselled about their options: ECV, planned cesarean birth, or planned vaginal birth. If women declined ECV or if it was unsuccessful, they were offered elective cesarean delivery from 39 weeks of gestation. The population was divided into 2 study groups: Group 1 (women who were offered and accepted a routine third trimester scan) and Group 2 (women who received standard antenatal care in line with national guidance, without a routine third trimester scan).

## POCUS cohort

The POCUS cohort included pregnant women from NNUH where a policy of routine POCUS at the 36-week antenatal visit was fully adopted from November 2020 following stage-wise implementation in 2016. The POCUS is performed by a midwife using Vscan Air (GE Healthcare). NNUH is a large teaching hospital with approximately 6,000 births per year, and approximately 250 midwives working across the hospital and community. We included 2 groups: a historical cohort of women who received routine care—abdominal palpation and referral for selective ultrasound on clinical suspicion of breech presentation (2015) and those who had POCUS at the 36- to 37-week visit (November 2020 to 2021). Through 2016 to November 2020, POCUS was variably used, either on the labour ward or via referral from community midwives, on clinical suspicion of noncephalic presentation, and these women were not included in this study.

## Training of midwives for POCUS cohort

The midwives in NNUH underwent a structured 3-month training programme. The workshops consisted of daily handheld scanning sessions with an hour of dedicated lectures. The theoretical lectures were followed by practice on consenting women in the antenatal ward. All the trainee midwives maintained a competency logbook, detailing both successful and unsuccessful cases. Following the initial workshops, "midwife champions" were identified who were deemed competent or held other ultrasound qualifications and were suitable for cascade training. POCUS training was a part of preceptor ship training of newly qualified midwives, while midwives working in nonpermanent roles were supported and advised to work with one of the champions.

The primary outcome was the proportion of all term breech presentations that were undiagnosed. Undiagnosed breech presentation was defined as follows: (a) breech presentation after the onset of labour or rupture of membranes at term; and (b) breech presentation diagnosed immediately before commencing induction of labour. The secondary outcomes included mode of birth, gestational age at birth, birth weight, incidence of emergency cesarean section, and the following neonatal adverse outcomes: Apgar score <7 at 5 minutes, unexpected neonatal unit (NNU) admission, hypoxic ischemic encephalopathy (HIE) 1 to 3, and perinatal mortality (includes stillbirths and early neonatal deaths).

Women with multiple pregnancies, preterm birth <37 weeks, and congenital abnormalities were excluded. Pregnancies undergoing planned cesarean section for breech presentation

were excluded from the analysis of the study outcomes, except for the neonatal outcomes. Maternal demographic characteristics, antenatal, intrapartum, and perinatal data were extracted from Euroking E3 maternity information system and Viewpoint database (View-Point 5.6.8.428, ViewPoint Bildverarbeitung GmbH, Weßling, Germany). Routinely collected clinical data were collated from electronic health records and were deemed not to require ethics approval or signed patient consent as per the Health Research Authority (HRA) decision tool.

## Statistical analysis

Descriptive variables were compared with Wilcoxon-signed rank test, $t$ test, or chi-squared test, where appropriate. An adequately powered analysis is not practically feasible due to rarity of adverse outcomes following breech delivery. Therefore, we employed a Bayesian approach using informative priors from a previous similar study; updating their estimates (prior) with our own data (likelihood) [18]. The association of undiagnosed breech presentation at birth with adverse perinatal outcomes was analyzed with Bayesian log-binomial regression models and reported as RR (risk ratios) with credible intervals (CrI). Informative priors ($N{\sim}\mu$, $\sigma$) for population mean were derived from Salim and colleagues and a weakly informative prior (Student $t$, df = 3) for model intercept. Prior parameters were estimated by using the log-risk ratios and log-confidence intervals from Salim and colleagues, and in case an effect could not be estimated in the original study due to a no-event situation, we added a single event to the corresponding group and reestimated the risk ratios. Two Markov chains were run for 1,500 iterations after an initial 500 burn-in period. Posterior probabilities were calculated using the probability density function of normal distribution. A sensitivity analysis using flat priors (noninformative) was also undertaken to investigate the weight of informative prior on the posterior density. Number needed to treat for important outcomes was calculated using current population numbers without incorporating external data. Convergence was checked with trace plots. All analyses were conducted using R for Statistical Software (v.4.2.0) using "brms" and "its.analysis" packages [25,26]. This study is reported as per the Strengthening the Reporting of Observational Studies in Epidemiology (STROBE) guideline (S1 STROBE Checklist).

## Results

### Study cohorts

In the SGH cohort, there were 24,128 singleton pregnancies during the eligibility period, of which 16,777 births were before the introduction of universal third trimester ultrasound scan and 7,351 after. Baseline characteristics of included pregnancies are presented in Table 1. Women who gave birth before universal ultrasound scan were significantly younger (33.2 versus 35.7 years, $p < 0.001$), had similar BMI (25.6 versus 25.7 kg/m$^2$, $p = 0.194$) and multiparity rate (49.6% [8,316/16,777] versus 49.2% [3,617/7,351], $p = 0.612$) compared to those who gave birth after. There was a slight drop in the proportion of births that were in women from black, Asian, and minority ethnic (BAME) background (39.3% [6,588/16,777] versus 37.9% [2,785/7,351], $p = 0.044$). The index of multiple deprivation (IMD) quintiles were similar between the 2 epochs ($p > 0.05$ for all quintiles; Table 1), as was the total number of breech presentations at the time of birth (3.4% [578/16,777] versus 3.4% [251/7,351], $p = 0.953$), including all diagnosed and undiagnosed cases. A comparison of the baseline characteristics, as well as the gestational age at delivery in weeks and mode of birth of pregnancies with breech presentation at birth in the study epochs before and after the introduction of universal 36-week ultrasound scan is shown in Table 2. Pregnancies with breech presentation at term were significantly more likely to be delivered by elective cesarean section (76.9% [193/251] versus 60.7% [351/

**Table 1. Baseline characteristics of pregnancies in the study epochs before and after the introduction of universal 36-week ultrasound scan.**

| Variables | Before universal 36-week scan (n = 16,777) | After universal 36-week scan (n = 7,351) | P value |
|---|---|---|---|
| Maternal age in years | 33.2 ± 5.18 | 35.7 ± 5.39 | <0.001 |
| BMI, kg/m² | 25.6 ± 5.35 | 25.7 ± 5.36 | 0.194 |
| Parous, n (%) | 8,316 (49.6) | 3,617 (49.2) | 0.612 |
| Ethnicity, n (%) | | | |
| Caucasian | 8,014 (47.8) | 3,463 (47.1) | 0.353 |
| BAME | 6,588 (39.3) | 2,785 (37.9) | 0.044 |
| Not stated | 2,175 (12.9) | 1,103 (15.0) | <0.001 |
| IMD quintile, n (%) | | | |
| First | 1,056 (6.3) | 423 (5.8) | 0.114 |
| Second | 4,323 (25.8) | 1,874 (25.5) | 0.665 |
| Third | 4,989 (29.7) | 2,278 (31.0) | 0.052 |
| Fourth | 3,740 (22.3) | 1,664 (22.6) | 0.566 |
| Fifth | 2,626 (15.6) | 1,087 (14.8) | 0.090 |
| Missing | 43 (0.3) | 25 (0.3) | 0.318 |
| All breech presentations, n (%) | 578 (3.4) | 251 (3.4) | 0.953 |
| Gestational age at delivery in weeks | 39.8 ± 1.13 | 39.9 ± 1.18 | <0.001 |
| Mode of birth, n (%) | | | |
| Elective cesarean | 2,019 (12.0) | 959 (13.0) | 0.029 |
| Emergency cesarean | 2,169 (12.9) | 845 (11.5) | 0.002 |
| Vaginal breech, any | 49 (0.29) | 15 (0.20) | 0.276 |
| Vaginal vertex, spontaneous | 9,606 (57.3) | 4,383 (59.6) | <0.001 |
| Vaginal vertex, operative | 2,934 (17.5) | 1,149 (15.6) | <0.001 |

Comparison between the 2 groups was performed using Wilcoxon-signed rank test, *t* test (for continuous variables), or chi-squared test (for binary or categorical variables), where appropriate.

BAME, black, Asian, and minority ethnic; BMI, body mass index; IMD, index of multiple deprivation.

278], $p < 0.001$) after compared to before the implementation of the universal 36-week ultrasound scan. Emergency cesarean section was lower (17.1% [43/251] versus 30.8% [178/578], $p < 0.001$) after compared to before the implementation of the universal 36-week ultrasound scan. A similar trend was noted for vaginal breech delivery (Table 2). The gestational age at birth was 39.1 weeks in both groups with a mean difference of 1 day. Although the difference was statistically significant, it would be deemed clinically inconsequential.

The percentage of all term breech presentations that were undiagnosed was 14.2% (82/578) before and 2.8% (7/251) after the implementation of universal screening ($p < 0.001$) (Table 3). The rate of elective cesarean delivery was higher during the universal scan epoch (13.0% [959/7,351] versus 12.0% [2,019/16,777], $p = 0.029$), while the emergency cesarean rate was lower (12.9% [2,169/16,777] versus 11.5% [845/7,351], $p = 0.029$) compared to the previous epoch. The total number of vaginal breech births reduced from 29 per 10,000 births to 20 per 10,000 births, but this difference did not reach statistical significance ($p = 0.276$). A regression discontinuity (interrupted time series) analysis showed significant variation in the number of undiagnosed breech presentations between 2 epochs (before and after universal ultrasound, $p < 0.001$) (Fig 1).

In the NNUH cohort, there were 9,694 singleton births during the eligibility period, of which 5,119 births were before the initiation of POCUS screening and 4,575 births after the complete implementation of POCUS. Women who gave birth before POCUS were significantly older (34.6 versus 31.6 years, $p < 0.001$) and had a lower BMI (25.6 versus 26.5 kg/m²,

**Table 2. Baseline characteristics of pregnancies with breech presentation at birth in the study epochs before and after the introduction of universal 36-week ultrasound scan.**

| Variable | After universal 36-week scan ($n$ = 251) | Before universal 36-week scan ($n$ = 578) | *P* value |
|---|---|---|---|
| Maternal age in years, median (IQR) | 35.0 (31.0–38.0) | 37.0 (33.2–40.0) | <0.001 |
| BMI in kg/m$^2$, median (IQR) | 24.0 (22.0–28.0) | 24.0 (22.0–28.0) | 0.616 |
| Parous, n (%) | 93 (37.1) | 209 (36.2) | 0.867 |
| Maternal ethnicity, n (%) | | | 0.177 |
| Caucasian | 127 (50.6) | 316 (54.7) | |
| Black | 15 (6.0) | 37 (6.4) | |
| Asian | 34 (13.5) | 77 (13.3) | |
| Mixed | 12 (4.8) | 11 (1.9) | |
| Any other ethnic group | 22 (8.8) | 61 (10.6) | |
| Not stated | 41 (16.3) | 76 (13.1) | |
| IMD 10 groups, n (%) | | | 0.755 |
| (0,2] | 16 (6.4) | 29 (5.0) | |
| (2,4] | 64 (25.6) | 135 (23.4) | |
| (4,6] | 78 (31.2) | 178 (30.8) | |
| (6,8] | 53 (21.2) | 128 (22.2) | |
| (8,10] | 39 (15.6) | 107 (18.5) | |
| Gestational age at delivery in weeks, median (IQR) | 39.1 (38.7 to 39.5) | 39.1 (38.7 to 39.9) | 0.006 |
| Mode of birth[#], n (%) | | | <0.001 |
| Breech extraction | 2 (0.8) | 13 (2.2) | |
| Breech, forceps to aftercoming head | 2 (0.8) | 4 (0.7) | |
| Elective cesarean section | 193 (76.9) | 351 (60.7) | |
| Emergency cesarean section | 43 (17.1) | 178 (30.8) | |
| Spontaneous breech | 11 (4.4) | 32 (5.5) | |

Comparison between the 2 groups was performed using Wilcoxon-signed rank test, *t* test, (for continuous variables), or chi-squared test (for binary or categorical variables), where appropriate.

[#]These categories are mutually exclusive.

BMI, body mass index; IMD, index of multiple deprivation.

$p < 0.001$) than those who gave birth after. The percentage of all term breech presentations that were undiagnosed was 16.2% (27/167) before and 3.5% (5/142) after the implementation of universal POCUS screening ($p < 0.001$) (Table 4).

**Perinatal outcomes.** We analysed the SGH cohort using Bayesian regression analysis with both flat (noninformative) and informative priors (Using data from Salim and colleagues) [18]. Regression with informative priors showed the percentage of all term breech presentations that were undiagnosed was 71% lower after the implementation of universal ultrasound (RR, 0.29; 95% CrI 0.20, 0.38) with a posterior probability greater than 99.9% (Table 3). Among the pregnancies with breech presentation, there was also a very high probability (>99.9%) of reduced rate of low Apgar score (<7) at 5 minutes by 77% (RR, 0.23; 95% CrI 0.14, 0.38). There was moderate to high probability (posterior probability: 89.5% and 85.1%, respectively) of a reduction of HIE (RR, 0.32; 95% CrI 0.05, 1.77) and extended perinatal mortality rates (RR, 0.21; 95% CrI 0.01, 3.00). Analysis using flat priors (noninformative) also showed that the percentage of all term breech presentations that were undiagnosed was 74% lower (RR, 0.26; 95% CrI 0.10, 0.59) with very high posterior probability of 99.8%. The reduction in low Apgar scores was also observed in flat prior analysis that corresponded to a 65% reduction (RR, 0.35; 95% CrI 0.06, 1.42) with a moderate to high probability (89.8%). The

**Table 3. Perinatal outcomes of breech presentations in the study epochs before and after the introduction of universal 36-week ultrasound scan in St George's Hospital.**

| Outcomes | Before universal ultrasound scan (*n* = 578) | After universal ultrasound scan (*n* = 251) | RR (95% CrI), flat prior* | Posterior probability of adverse event reduction | RR (95% CrI), informative prior† | Posterior probability of adverse event reduction |
|---|---|---|---|---|---|---|
| Term breech presentations that were undiagnosed, n (%) | 82 (14.2) | 7 (2.8) | 0.26 (0.10, 0.59) | 99.8% | 0.29 (0.20, 0.38) | >99.9% |
| NNU admission, n (%) | 11 (1.9) | 3 (1.2) | 0.55 (0.12, 1.98) | 79.7% | 0.84 (0.55, 1.28) | 78.0% |
| HIE, n (%) | 2 (0.3) | 1 (0.4) | 1.19 (0.10, 25.1) | 44.8% | 0.32 (0.05, 1.77) | 89.5% |
| Apgar <7 at 5 minutes, n (%) | 11 (1.9) | 2 (0.8) | 0.35 (0.06, 1.42) | 89.8% | 0.23 (0.14, 0.38) | >99.9% |
| Extended perinatal mortality, n (%) | 2 (0.3) | 0 (0.0) | NE | – | 0.21 (0.01, 3.00) | 85.1% |

*Weakly informative priors (N~μ,σ) for population mean and (Student t, df = 3) model intercept.

†Informative priors (N~μ,σ) for population mean was derived from Salim and colleagues and a weakly informative prior (Student t, df = 3) for model intercept.

CrI, credible interval; HIE, hypoxic ischemic encephalopathy; NE, not estimable; NNU, neonatal unit; RR, relative risk.

number needed to scan to prevent one case of undiagnosed breech presentation was 255 (95% CI: 192 to 376).

We analysed the NNUH cohort using the same methods. Using informative priors, the proportion of all term breech presentations that were undiagnosed was 69% lower after the initiation of universal POCUS (RR, 0.31; 95% CrI 0.21, 0.45) with a posterior probability greater of 99.9% (Table 4). There was also a very high probability (99.5%) of a reduced rate of low Apgar score (<7) at 5 minutes by 40% (RR, 0.60; 95% CrI 0.39, 0.88). Flat prior analysis also showed that undiagnosed breech presentation was lower by 80% (RR, 0.20; 95% CrI: 0.07, 0.51) with very high posterior probability of 99.9%. No inference could be made for HIE or extended perinatal mortality as there were no events in either period.

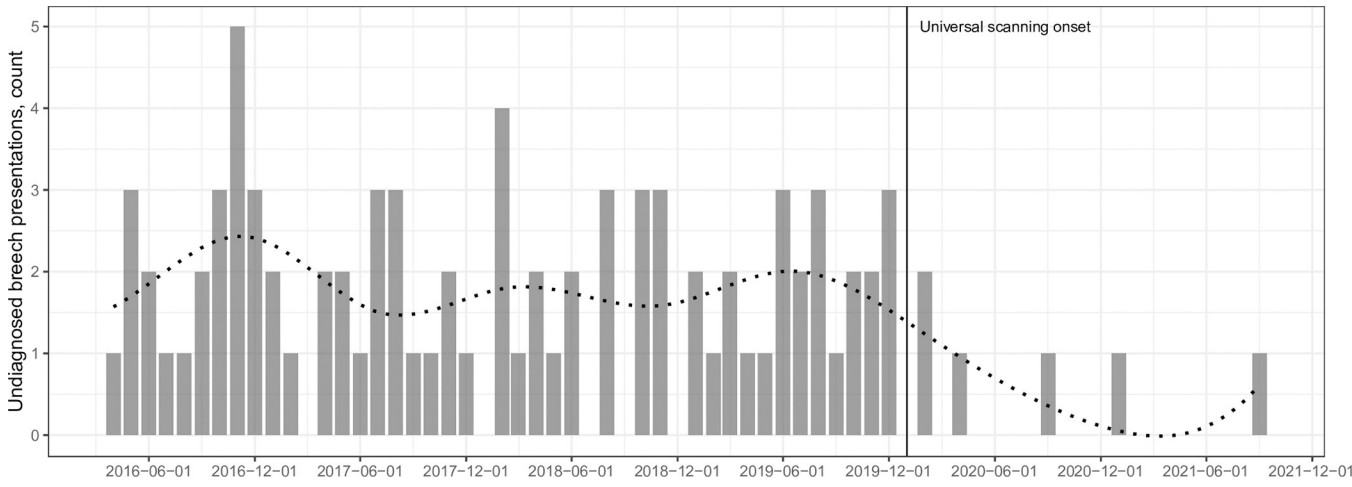

**Fig 1. The number of term breech presentations that were undiagnosed during the study period in St. George's University Hospital.** Dotted line represents the fitted regression curve with splines.

**Table 4. Perinatal outcomes of breech presentations in the study epochs before and after the introduction of universal POCUS at 36 weeks in in Norfolk Norwich University Hospital.**

| Outcomes | Before universal POCUS (n = 167) | After universal POCUS (n = 142) | RR (95% CrI), flat prior* | Posterior probability of adverse event reduction | RR (95% CrI), informative prior† | Posterior probability of adverse event reduction |
|---|---|---|---|---|---|---|
| Term breech presentations that were undiagnosed, n (%) | 27 (16.2) | 5 (3.5) | 0.20 (0.07, 0.51) | 99.9% | 0.31 (0.21, 0.45) | >99.9% |
| NNU admission, n (%) | 5 (3.0) | 3 (2.1) | 0.53 (0.11, 2.40) | 78.4% | 0.84 (0.54, 1.34) | 77.0% |
| HIE, n (%) | 0 (0.0) | 0 (0.0) | NE | – | NE | – |
| Apgar <7 at 5 minutes, n (%) | 3 (1.8) | 2 (1.4) | 0.72 (0.10, 4.30) | 63.0% | 0.60 (0.39, 0.88) | 99.5% |
| Extended perinatal mortality, n (%) | 0 (0.0) | 0 (0.0) | NE | – | NE | – |

*Weakly informative priors ($N\sim\mu, \sigma$) for population mean and (Student $t$, df = 3) model intercept.

†Informative priors ($N\sim\mu, \sigma$) for population mean was derived from Salim and colleagues and a weakly informative prior (Student $t$, df = 3) for model intercept.

CrI, credible interval; HIE, hypoxic ischemic encephalopathy; NE, not estimable; NNU, neonatal unit; POCUS, point-of-care ultrasound; RR, relative risk.

## Discussion

In our study, use of a policy of either routine third trimester scan or routine third trimester POCUS was associated with a significant reduction in the proportion of all breech presentations that were undiagnosed at term, when compared to standard antenatal care. Short-term adverse perinatal outcomes, including NNU admission and low Apgar scores, were significantly lower for the pregnancies with diagnosed breech presentation at term following a policy for screening by either routine third trimester scan or POCUS. Previous studies were unable to study perinatal outcomes due to their small numbers.

Our cohorts are derived from real-world data from 2 large teaching hospitals in the UK. Much of the previously reported literature on use of third trimester ultrasound for diagnosis of breech presentation is from research settings with a dedicated breech clinic and available expertise and skills for manoeuvres like ECV and vaginal breech births. Conclusions from research settings may not be generalizable to clinical settings and may be prone to bias. Furthermore, our study is the first to compare the impact of POCUS with routine antenatal care for diagnosis of fetal presentation. Routine ultrasound scan is effective at reducing the proportion of all term breech presentations that were undiagnosed, but the clinical impact of this change is hard to assess owing to the rarity of adverse outcomes [18]. We employed a Bayesian approach using both informative priors from similar studies and flat priors as a sensitivity analysis that allowed us to estimate the effect of universal ultrasound in probabilistic terms without depending on $P$ values.

There are some limitations to our study. Firstly, we did not have reliable data on ECV for both our cohorts. The universal scan might have implications, not just for babies that were breech at birth (e.g., ECV could be performed, which could lead to not being breech and therefore not being included in the outcomes, or some other benefit, or indeed, theoretically, harm). Salim and colleagues included all babies diagnosed as breech. The method employed by Salim and colleagues also has drawbacks as it did not include those undiagnosed before the universal scan. Nevertheless, it is unlikely to have had a substantial impact on our results given the low acceptance and variable success rates. This is reflected in the almost identical incidence of overall (undiagnosed and diagnosed) breech presentation before and after screening. Salim and colleagues also reported no difference in the rates of overall breech presentations despite systematic use of ECV with acceptance rates of as high as 80% [18]. Secondly, the number of

adverse neonatal outcomes such as extended perinatal mortality and HIE were not sufficient to estimate an effect in the NNUH cohort. Finally, the maternity records at NNUH were uploaded on electronic database only from April 2015. Therefore, reliable data on demographic parameters like BMI, ethnicity, and IMD were not available for the first quarter of 2015. These factors, however, are unlikely to influence the results.

Our findings of a reduction in the proportion of all term breech presentations that were undiagnosed at term after implementation of routine third trimester scan resonates with those of Salim and colleagues, who reported a reduction from 22.3% to 4.7% following the introduction of universal third trimester scan, compared to standard antenatal care [18]. Yet there are no published data from the UK on the impact of routine POCUS on the reduction of the percentage of all term breech presentations that were undiagnosed at term. Observational studies from Kenya [19], Uganda [27], and Guatemala [28] have reported that midwives who underwent focused basic obstetric ultrasound training for 3 to 8 weeks were able to identify fetal presentation with high sensitivity and specificity. The proportion of all term breech presentations that were undiagnosed at term, however, could not be eliminated in both cohorts, with 7 and 5 such cases in the routine third trimester scan and POCUS cohorts, respectively. Most of these cases were a consequence of spontaneous version to breech from cephalic presentation in multiparous women. Salim and colleagues also described spontaneous version to breech in multiparous women (76% of cases of undiagnosed breech) in their cohort. Wastlund and colleagues reported in their prospective cohort of 3,879 women that a policy of universal third trimester scan virtually eliminated undiagnosed breech presentations in labour [16]. It should, however, be noted that their cohort comprised of nulliparous women only in a strict research setting.

We also noted a significant improvement in short-term neonatal outcomes including low Apgar scores at 5 minutes and NNU admission. Salim and colleague demonstrated a nonsignificant improvement in short-term neonatal outcomes [18]. Although we were unable to demonstrate an effect on outcomes such as HIE and neonatal mortality, observational data from low-resource settings report a reduction in neonatal mortality when women were referred in a timely manner for fetal malpresentation [29].

Accurate knowledge of fetal presentation at term is crucial for optimal antenatal and intrapartum care. Both routine third trimester scan by a sonographer/clinician or use of POCUS by trained midwives can achieve this objective. Although evidence suggests that a planned breech vaginal birth may be offered after careful case selection and counselling, a large proportion of maternity units in the UK and worldwide lack skilled providers for vaginal breech births. Antenatal identification of breech presentation would allow healthcare providers to offer unbiased information such that pregnant women feel empowered to make an informed decision and have a positive birth experience. A meeting of the UK National Screening Committee (NSC) in March 2021 acknowledged that ultrasound for fetal presentation appears promising; however, the committee recommended that further work on screening for fetal presentation could not be commissioned at that time. The NSC agreed to add screening for fetal presentation to the recommendations list for reconsideration in 3 years' time if significant evidence evolves in the interim [30]. Our findings add to that evidence base. A cost-effectiveness analysis study conducted in the UK showed that universal third trimester ultrasound would "virtually eliminate" the proportion of all term breech presentations that were undiagnosed and would be cost-effective if fetal presentation could be assessed at £19.80 pounds per woman or less [16]. A National Institute for Health Research (NIHR) Health Technology Assessment (HTA) review has suggested that handheld portable ultrasound can readily close this gap as a low-cost device that antenatal care providers like midwives could use for fetal presentation with minimal training [31]. The major obstacles to routine third trimester scan policy include the costs incurred, whereas a policy of using POCUS in community clinics and the labour

ward by healthcare providers, after a short period of training, appears to be as effective as a policy of routine third trimester formal departmental ultrasound. Implementation of POCUS in the community for fetal presentation would also curtail indirect costs by reduction in referrals for facility-based ultrasound based on clinical suspicion, apart from also instilling a sense of empowerment among the care providers and satisfaction among pregnant women. The policy of POCUS was acceptable to pregnant women in our cohort who wanted to avoid nonessential hospital visits during the COVID-19 pandemic. A potential pitfall of the portable ultrasound cited when used in low-resource settings was dependence on internet coverage, which is unlikely to be a deterrent in the UK. Nonetheless, regular audits, ongoing training, and quality improvement measures should be in place to support community healthcare providers to ensure safe practice.

Our data suggest that a policy of either third trimester ultrasound or POCUS by healthcare providers could be effective in reducing the proportion of all term breech presentations that were undiagnosed at birth with an associated improvement in neonatal outcomes.

## Supporting information

**S1 STROBE Checklist. STROBE checklist.**
(DOCX)

## Author Contributions

**Conceptualization:** Smriti Prasad, Erkan Kalafat, Pam Sizer, Francoise Harlow, Asma Khalil.

**Data curation:** Samantha Knights, Smriti Prasad, Anahita Dadali.

**Formal analysis:** Erkan Kalafat.

**Methodology:** Erkan Kalafat, Asma Khalil.

**Supervision:** Erkan Kalafat, Asma Khalil.

**Validation:** Erkan Kalafat.

**Writing – original draft:** Smriti Prasad, Erkan Kalafat, Asma Khalil.

**Writing – review & editing:** Smriti Prasad, Erkan Kalafat, Francoise Harlow, Asma Khalil.

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
