## [Editor Report · Decision Letter 0]

25 Aug 2022

Dear Dr Khalil, 

Thank you for submitting your manuscript entitled "Impact of Point of Care Ultrasound and routine third trimester ultrasound on undiagnosed breech presentation and perinatal outcomes" for consideration by PLOS Medicine.

Your manuscript has now been evaluated by the PLOS Medicine editorial staff and I am writing to let you know that we would like to send your submission out for external peer review.

Please re-submit your manuscript within two working days, i.e. by Aug 29 2022 11:59PM.

Kind regards,

Beryne Odeny

PLOS Medicine

---

## [Decision Letter · Decision Letter 1]

16 Nov 2022

Dear Dr. Khalil,

Thank you very much for submitting your manuscript "Impact of Point of Care Ultrasound and routine third trimester ultrasound on undiagnosed breech presentation and perinatal outcomes" (PMEDICINE-D-22-02818R1) for consideration at PLOS Medicine. 

Your paper was evaluated by a senior editor and discussed among all the editors here. It was also sent to independent reviewers, including a statistical reviewer. The reviews are appended at the bottom of this email and any accompanying reviewer attachments can be seen via the link below:

[LINK]

In light of these reviews, I am afraid that we will not be able to accept the manuscript for publication in the journal in its current form, but we would like to consider a revised version that addresses the reviewers' and editors' comments. Obviously we cannot make any decision about publication until we have seen the revised manuscript and your response, and we plan to seek re-review by one or more of the reviewers. 

We expect to receive your revised manuscript by Dec 07 2022 11:59PM. Please email us (plosmedicine@plos.org) if you have any questions or concerns.

We look forward to receiving your revised manuscript. 

Sincerely,

Pippa

Philippa Dodd MBBS MRCP PhD

PLOS Medicine

plosmedicine.org

GENERAL

Please respond to all editor and reviewer comments detailed below, in full.

Please ensure that the study is reported according to the STROBE guideline, and include the completed STROBE checklist as Supporting Information. Please add the following statement, or similar, to the Methods: "This study is reported as per the Strengthening the Reporting of Observational Studies in Epidemiology (STROBE) guideline (S1 Checklist)." The STROBE guideline can be found here: http://www.equator-network.org/reporting-guidelines/strobe/ When completing the checklist, please use section and paragraph numbers, rather than page (or line) numbers.

Throughout, please ensure that all abbreviation are defined at first use 

DATA AVAILABILITY STATEMENT

PLOS Medicine requires that the de-identified data underlying the specific results in a published article be made available, without restrictions on access, in a public repository or as Supporting Information at the time of article publication, provided it is legal and ethical to do so. Please see the policy at http://journals.plos.org/plosmedicine/s/data-availability and FAQs at http://journals.plos.org/plosmedicine/s/data-availability#loc-faqs-for-data-policy

The Data Availability Statement (DAS) requires revision. For each data source used in your study: 

TITLE

Please revise your title according to PLOS Medicine's style. Your title must be nondeclarative and not a question. It should begin with main concept if possible. "Effect of" should be used only if causality can be inferred, i.e., for an RCT. Please place the study design ("A randomized controlled trial," "A retrospective study," "A modelling study," etc.) in the subtitle (ie, after a colon).

ABSTRACT

Please structure your abstract using the PLOS Medicine headings (Background, Methods and Findings, Conclusions). Please combine the Methods and Findings sections into one section, “Methods and findings”.

* Abstract Background: 

Pleas provide the context of why the study is important. The final sentence should clearly state the study question.

* Abstract Methods and Findings: 

Please ensure that all numbers presented in the abstract are present and identical to numbers presented in the main manuscript text. 

Please include the study design, population and setting, number of participants, years during which the study took place, length of follow up, and main outcome measures. 

Please quantify the main results with 95% CIs and p values. 

Please include the important dependent variables that are adjusted for in the analyses. 

Please include the actual amounts and/or absolute risk(s) of relevant outcomes (including NNT or NNH where appropriate), not just relative risks or correlation coefficients. (example for absolute risks: PMID: 28399126). 

Please ensure you clearly report numerators and denominators used to derive percentages

Please revise your statistical reporting such that all numerical values are clearly defined for the reader. Please define CI, RR etc at first use

Please include a summary of adverse events if these were assessed in the study. 

In the last sentence of the Abstract Methods and Findings section, please describe the main limitation(s) of the study's methodology.

* Abstract Conclusions: 

Please expand the conclusion, in -keeping with the below comments:

Please address the study implications without overreaching what can be concluded from the data; the phrase "In this study, we observed ..." may be useful. 

Please interpret the study based on the results presented in the abstract, emphasizing what is new without overstating your conclusions. 

Please avoid vague statements such as "these results have major implications for policy/clinical care". Mention only specific implications substantiated by the results. 

Please avoid assertions of primacy ("We report for the first time....")

AUTHOR SUMMARY

Please place this author summary, appropriately structured with bulleted point, under sub-headings (as detailed in our guidance)

METHODS and RESULTS

Please ensure you have clearly reported the number of participants dates of recruitment, and account for all methods used in your study.

Please define "lost to follow-up" as used in this study. Other reasons for exclusion should be defined.

Please define the length of follow up (eg, in mean, SD, and range).

Please indicate which, if any, factors have been adjusted for in your analyses

Did your study have a prospective protocol or analysis plan? Please state this (either way) early in the Methods section. 

For all observational studies, we ask that in the manuscript text you clearly indicate: 

(1) the specific hypotheses you intended to test, 

(2) the analytical methods by which you planned to test them, 

(3) the analyses you actually performed, and

(4) when reported analyses differ from those that were planned, transparent explanations for differences that affect the reliability of the study's results. If a reported analysis was performed based on an interesting but unanticipated pattern in the data, please be clear that the analysis was data-driven If analyses were performed in response to the peer review process, please clearly indicate this.

Throughout, we politely request that you be more nuanced in respect to statistical reporting. 

Please provide the actual numbers of events for the outcomes, not just summary statistics or ORs. 

Where you report percentages, please clearly report numerators and denominators used to derive them. 

Please quantify main results with 95% CIs as well as p-values. 

Where p-values are reported please detail the statistical tests used to determine them.

TABLES

Please include appropriate captions which clearly detail the content of the tables, and which define numerical values contained within. Please ensure all abbreviations are also clearly defined for each table e.g., RR, CI, SGHG, NNUH, NE

Are any factors adjusted for in the analyses presented here? Please detail such factors in the appropriate caption and where adjusted analyses are presented, please also include the unadjusted analyses to help facilitate transparency of observational data reporting.

Please quantify main results reported in the tables with 95% CIs and p-values.

Please indicate in the captions the statistical tests used to determine p-values

Table 2 – suggest removing the second header row between the cohorts – I appreciate that this is to detail the difference in n numbers across the different cohorts but a lot of information is repeated, perhaps unnecessarily, to satisfy that specific requirement. Please revise the presentation of this table to improve accessibility to the reader

Please indicate where in the manuscript the tables should be placed

FIGURES

Please consider using a color palate that is accessible to those with color blindness (i.e., avoids the uses of red and/or green)

Please indicate where in the manuscript the figures should be placed

DISCUSSION

Please remove all subheadings from the discussion such that it reads as one continuous piece of prose ending in a one paragraph conclusion.

Please remove COI statement, funding statement and ethics statement from the end of the main manuscript and include only in the manuscript submission form.

Please remove the patient consent statement and incorporate with your ethics statement. 

REFERENCES

For in-text reference callouts, citations should be in square brackets, and preceding punctuation as follows: “…[1,2,3,4].” Please not the absence of spaces where more than one citation is reported. 

In the reference list, where more than 6 authors contribute to a study please list the first 6 only followed by et al.

Please see our website for other reference guidelines https://journals.plos.org/plosmedicine/s/submission-guidelines#loc-references

Journal name abbreviations should be those found in the National Center for Biotechnology Information (NCBI) databases.

Comments from the reviewers:

Reviewer #1: 

This study assesses the 'impact' of a universal third trimester ultrasound on breech presentation. It concludes, using 2 'before and after' cohorts from 2 different units introducing 2 different types of 3rd trimester scans, that undiagnosed breech presentation is significantly reduced, that the incidence of breech presentation stays the same, and that the incidence of low Apgar scores is significantly reduced. 

The first two findings subsequent to a universal scan have been shown before in a single unit cohort previously published in Plos Medicine (salim et al) to which this papers refers, and from which the analysis takes 'informative priors'. The reduction in undiagnosed breech has been shown too, albeit in nulliparous women in research setting, also published in this journal (Wastlund et al) (also cited in this paper). The reduction in Apgar scores was not 'significant' in the Salim study, but with this, it is. That this is so when more data is used should not be considered a surprise: antenatal diagnosis allows more caesarean sections (I note overall numbers not changed but no analysis is specifically presented on cs rates for breech babies- see point below) for breech babies: caesarean sections for breech improve short term outcomes and, according to the term breech trial and huge cohorts from the Netherlands, death.

The methodology requires scrutiny by a statistical reviewer and I am not one. This means that I do not fully understand- and apologise for this- how, for instance, despite my point above, the paper concludes that 5 min Apgar scores <7 are reduced, when using a simple chi-squared test amalgamating numbers from Salim et al and from this paper I calculate a chi-square statistic of 2.2251; p= 0.14: i.e. still 'not significant' in lay terms.

Although I have outlined that antenatal diagnosis will have benefits, the abstract suggests to the uninitiated reader like me that the paper actually demonstrates a reduction in HIE: 'There was moderate to high probability (posterior probability: 89.5% and 85.1%, respectively) of a reduction of hypoxic ischemic encephalopathy (HIE) (RR: 0.23, 95% CI: 0.14-0.38)… '. Yet this is not the case, the numbers simply being inadequate for this: table 2 shows just 2 before and 1 after the scan was introduced. Without fully understanding the methodology this seems disingenuous.

Another issue with this paper is the loss of the 'real world' (what happens in a unit when…) benefits to an impact/ 'changes subsequent to' paper. This is because of there being 2 different units, the intervention was slightly different (proper scan and presentation scan) and ECV was not documented/ discussed. This means it potentially has less relevance for a 'unit' and is more of a 'poor man's' (person's) RCT.

I remain surprised at all the research into a universal 'presentation only' scan. I think we know the answer- most breeches become diagnosed. This affects only about 5% of the population. Given the (accepted) benefits of elective caesarean for breech babies have probably been exaggerated (RCOG Greentop Guideline 2017) the overall effect on pregnancy safety is going to be small. And of course performing caesarean sections on all breech babies leads to a loss of skills and therefore probably increased risk for the small (in this study and that of Salim et al) but persistent breeches that still slip through. I make this comment because this pertains to the relevance of this- and other studies such as Salim et al- for overall pregnancy safety; in contrast, a 'proper' scan that could affect the other 95% as well is the one that needs the research attention. 

I would like to see data on how many third trimester scans were actually performed- before and after. 

No table data on gestation at birth or mode of birth is presented for the babies who were actually breech.

The lack of ECV data is acknowledged as a limitation but inadequately so. The Salim study they quote was not a 'research setting' and its data and discussion of ECV was comprehensive. This lack also causes the problem below: 

By looking at outcomes in breech babies only at birth (this is not 100% clear at least in the table) an assumption is being made that the universal scan could not alter outcomes for other babies. It could- not just for babies that were breech at birth (eg ECV could be performed which could lead to not being breech and therefore not being included in the outcomes, or some other benefit, or indeed, theoretically, harm). Salim et al 'get around' this by including all babies diagnosed as breech. The method employed by Salim et al has drawbacks too (it did not include those undiagnosed before the universal scan), but the method used here should at least be justified and discussed as a limitation. 

The pretty Figs 1 and 2 add little useful information bar perhaps demonstrating causality better re undiagnosed breech presentation- but this should be obvious by now. 

In spite of my largely minor reservations, if this stands up to statistical review, this paper cleverly demonstrates one of the benefits to spreading the use of ultrasound in pregnancy. 

Reviewer #2: This was a clinically meaningful study, well-written and generally easy to follow. The conclusions are clear and compelling. The role of midwives in delivering POCUS is especially unique and helpful given the potential for this model to be introduced more widely in clinical settings globally. The use of probabilities in the results is confusing to follow, the risk reduction is much easier to understand clinically, however these results are presented in the context of the analysis conducted.

The only suggested changes relate to the interpretation of findings. In the Discussion, I think it is important to clearly mention a) the role of mode of birth in the risk of neonatal adverse outcomes (ie what proportion of women with breech babies proceeded to CS and thus what was the impact of MOB on outcomes, if any, assuming that CS eliminates most of the breech-related risks of vaginal breech birth) and b) the role of parity or the proportion of multiparous women in each group.

Reviewer #3: 1. Line 120-122. "For both centers, pregnancies were grouped according to whether they received routine third-trimester scan (SGH) or POCUS (MNUH)." For the analysis, did the author include all the pregnancies or just include those received routine third-trimester scan at SGH or PROCUS at MNUH?

2. What's the reason for only including SGH cohort was in Table 1/Figure 1 ? How about the NNUH cohort?

3. As the log-binomial regression is used to analyze the association between undiagnosed breach presentation at birth and adverse perinatal outcomes, should the outcome be OR (odds ratio) rather than RR (relative risk?)

Reviewer #4: Thanks for allowing me to review your very interesting manuscript. I agree that the data presented show an impressive reduction in the chance of an undiagnosed breech presentation at term after both routine ultrasound screening and POCUS, carried out in two different hospitals. It would, of course, be a more rigorous study design to compare routine antenatal care with both routine ultrasound and POCUS in the same setting, or to have randomised rather than use a before and after design, but based on your data both methods appear to reduce the percentage of breech presentations that are undiagnosed at the start of labour or membrane rupture. I have a few minor points to make which could lead to some minor revisions:

1. I found the phrase 'the rate of undiagnosed breech' a bit confusing as it suggested to me that, for example, 14.2% of babies were presenting in labour as undiagnosed breeches. It might be better to say the 'percentage of all term breech presentations that were undiagnosed was 14.2%.....'

2. The statement that there was no significant change in the rate of vaginal breech birth (and the figures are given of 10 per 10,000, which was reducing to 5 per 10,000 after the scanning interventions) is somewhat surprising given the very significant change (reduction) in the percentage of undiagnosed breech presentations. Are you able to explain that? 

3. These data would suggest to me that prior to the intervention most breech babies were born by c/section, so it is then a little hard to understand how such an improvement in perinatal outcome has been achieved. I realise that the outcome data were secondary outcomes but the paper seems to argue that there is a high rate of morbidity/mortality associated with vaginal breech birth, yet the improvement in perinatal outcomes has occurred without much change in the rate. Indeed, given the numbers quoted the numbers of term vaginal births at SGH during the study period would have reduced from about 17 in the pre-ultrasound period to 4 after. To have seen a measurable reduction in the rate of low Apgar scores and rates of HIE would be remarkable, and would imply that outcomes for vaginal breeches were very poor in the pre-ultrasound period. Can you comment further on this? 

3. A minor point is that data is a plural noun and is sometimes used as a singular one (e.g. page 172)

4. Finally, I was left uncertain as to whether or not you would advocate a policy of routine ultrasound or POCUS or a hybrid of the two depending on risk? Or do you have plans to compare the two approaches in the same setting?

[LINK]

---

## [Decision Letter · Decision Letter 2]

19 Jan 2023

Dear Dr. Khalil,

Thank you very much for re-submitting your manuscript "Impact of Point of Care Ultrasound and routine third trimester ultrasound on undiagnosed breech presentation and perinatal outcomes: a retrospective study" (PMEDICINE-D-22-02818R2) for review by PLOS Medicine.

I have discussed the paper with my colleagues and the academic editor and it was also seen again by 4 reviewers. I am pleased to say that provided the remaining editorial and production issues are dealt with we are planning to accept the paper for publication in the journal.

[LINK]

We look forward to receiving the revised manuscript by Jan 26 2023 11:59PM.   

Sincerely,

Philippa Dodd, MBBS MRCP PhD

PLOS Medicine

plosmedicine.org

Requests from Editors:

GENERAL

Thank you for your detailed and considerate responses to previous editor and reviewer comments. Please see below for further comments that we require you respond to in full.

TITLE

Thank you for revising your title suggest “Impact of Point of Care Ultrasound and routine third trimester ultrasound on undiagnosed breech presentation and perinatal outcomes: an observational multi-centre cohort study” or something similar

ABSTRACT

Thank you for revising your abstract and for including p-values as well as 95% CIs. For each outcome measure reported we request that both p-values and 95% CIs are reported. If not please clearly state the reasons why not, for purposes of transparent data reporting (please remember that when reporting p-values to report as p<0.001 or where higher the exact p-value as p= 0.002, for example).

Suggest reporting as follows – for example, line 59 - (RR: 59 0.29, 95% CI [0.20–0.38], p<0.001) for optimal reader accessibility. 

Please note the use of parentheses around the CIs and the absence of the semi-colon. You may wish to consider separating upper and lower limits with commas rather than hyphens (as these can be confused with negative values) but we leave it to your discretion.

Please check and revise throughout, including in the main manuscript text and tables where relevant – please see below also

AUTHOR SUMMARY

Thank you for including an author summary which reads very nicely.

Line 100: we appreciate the explanation of your chosen methodological approach, suggest revising to ensure that the statement is interpretable to the more general reader – some may be unfamiliar with the concept of informative priors, for example

Line 103: “incidence of percentage…” suggest removing of percentage perhaps?

Line 104: suggest quantifying “drastically (a percentage will be adequate here)

Line 113: It would be helpful to elaborate on the reasons why assessing cost-effectiveness is necessary – i.e. for feasibility of implementation on a wider scale

METHODS and RESULTS

Please revise statistical reporting as suggested above

please quantify all reported percentages with numerators and denominators, for example line 271, “(12.9 vs 11.5%, p=0.029)” - please check and amend throughout

Please also report CIs where p-values are reported, if not please clearly state why not, for purposes of transparent data reporting

TABLES

Currently journal requirements stipulate that where 95% CIs are presented, including in tables, that p-values are also reported. Please include p-values but if not in your rebuttal letter, please clearly state the reasons why not, for transparency of data reporting

Table 3.a – suggest removing the SGH cohort row – perhaps a remnant of the previous version

DISCUSSION

Please remove the sub-heading conclusion. 

Line 404: given the observational nature of the study suggest tempering the language here slightly, “Our data suggest that a policy….could be effective in…” or something similar, for example

SOCIAL MEDIA

To help us extend the reach of your research, please provide any Twitter handle(s) that would be appropriate to tag, including your own, your co-authors’, your institution, funder, or lab. Please include these in the manuscript submission form when you re-submit your manuscript

Comments from Reviewers:

Reviewer #1: the authors have addressed all points satisfactorily

Reviewer #2: Thank you for your thorough attempts to rectify requested changes. The manuscript has been satisfactorily updated and improved.

Reviewer #3: I appreciate authors' effors in addressing the questions and revising the paper. I have no further comments.

Reviewer #4: Thanks for responding to my previous comments and to those of other reviewers. I have three relatively minor additional points to make about your revised manuscript;

1. Line 50 (Abstract). Can you confirm that the statement 'the rate of breech presentation in labour was consistent across all groups (3-4%)' is correct and does this refer to all births? It seems very high if the usual exclusions of preterm, multiples, congenital anomalies are applied.

2. Line 116 (Introduction). Similarly, is this figure of 3-4% used in the Introduction inclusive of all births, and does it exclude multiple pregnancies?

3. Line 175. I would think that 'Southwest London' is usually written as 'South West London' but as I am not a resident of London I am happy to be corrected.

[LINK]

---

## [Editor Report · Decision Letter 3]

7 Feb 2023

Dear Dr Khalil, 

On behalf of my colleagues and the Academic Editor, Dr. Anthea C Lindquist, I am pleased to inform you that we have agreed to publish your manuscript "Impact of Point of Care Ultrasound and routine third trimester ultrasound on undiagnosed breech presentation and perinatal outcomes: an observational multi-centre cohort study" (PMEDICINE-D-22-02818R3) in PLOS Medicine.

Prior to publication, on advice of the Academic Editor, whose comments are also included below, we require that you make the following final revisions:

This is an important study with direct clinical impact and will be of great interest to maternity clinicians globally.

The suggested revisions have generally been very appropriately addressed or appropriately rebutted. There are a few remaining issues to address:

Line 103 - wording doesn't make sense, needs to be clarified

Line 222 - 'excluding' should be 'excluded'

Line 226 - 'was' should be 'were' in keeping with the plural nature of data

Line 270-271 - reads as though the GA was the same (39.1 weeks) between groups but still significantly different - needs to be clarified, how can this differ?

Table 2 - mode of birth - needs to include in legend if all 3 breech vaginal birth categories (breech extraction, breech with forceps and spontaneous breech) were mutually exclusive or not? The same babies could all feasibly be included in all groups.

Line 317-318 - 'term' is used twice, only one is necessary

Line 328 - should read 'the first'

Line 331 - should read 'the rarity'

Line 349 - 'was not' should read 'were not' owing to plural data

Line 359-360 - 'term' is used twice, only needed once

Line 371 - remove 'the' from 'the neonatal'

PRESS

Thank you again for submitting to PLOS Medicine, it has been a pleasure handling your manuscript. We look forward to publishing your paper. 

Best wishes, 

Pippa

Philippa Dodd, MBBS MRCP PhD 

PLOS Medicine